# Synthesizing High b-Value Diffusion-Weighted Imaging of Gastric Cancer Using an Improved Vision Transformer CycleGAN

**DOI:** 10.3390/bioengineering11080805

**Published:** 2024-08-08

**Authors:** Can Hu, Congchao Bian, Ning Cao, Han Zhou, Bin Guo

**Affiliations:** 1School of Computer and Soft, Hohai University, Nanjing 211100, China; hucan@hhu.edu.cn (C.H.); congchaobian@hhu.edu.cn (C.B.); 2School of Electronic Science and Engineering, Nanjing University, Nanjing 210046, China; dg20230063@smail.nju.edu.cn; 3College of Computer and Information Engineering, Xinjiang Agricultural University, Urumqi 830052, China; gb@xjau.edu.cn

**Keywords:** gastric cancer, diffusion-weighted imaging, CycleGAN, vision transformer, medical image synthesis

## Abstract

Background: Diffusion-weighted imaging (DWI), a pivotal component of multiparametric magnetic resonance imaging (mpMRI), plays a pivotal role in the detection, diagnosis, and evaluation of gastric cancer. Despite its potential, DWI is often marred by substantial anatomical distortions and sensitivity artifacts, which can hinder its practical utility. Presently, enhancing DWI’s image quality necessitates reliance on cutting-edge hardware and extended scanning durations. The development of a rapid technique that optimally balances shortened acquisition time with improved image quality would have substantial clinical relevance. Objectives: This study aims to construct and evaluate the unsupervised learning framework called attention dual contrast vision transformer cyclegan (ADCVCGAN) for enhancing image quality and reducing scanning time in gastric DWI. Methods: The ADCVCGAN framework, proposed in this study, employs high b-value DWI (b = 1200 ${s/\mathrm{mm}}^{2}$) as a reference for generating synthetic b-value DWI (s-DWI) from acquired lower b-value DWI (a-DWI, b = 800 ${s/\mathrm{mm}}^{2}$). Specifically, ADCVCGAN incorporates an attention mechanism CBAM module into the CycleGAN generator to enhance feature extraction from the input a-DWI in both the channel and spatial dimensions. Subsequently, a vision transformer module, based on the U-net framework, is introduced to refine detailed features, aiming to produce s-DWI with image quality comparable to that of b-DWI. Finally, images from the source domain are added as negative samples to the discriminator, encouraging the discriminator to steer the generator towards synthesizing images distant from the source domain in the latent space, with the goal of generating more realistic s-DWI. The image quality of the s-DWI is quantitatively assessed using metrics such as the peak signal-to-noise ratio (PSNR), structural similarity index (SSIM), feature similarity index (FSIM), mean squared error (MSE), weighted peak signal-to-noise ratio (WPSNR), and weighted mean squared error (WMSE). Subjective evaluations of different DWI images were conducted using the Wilcoxon signed-rank test. The reproducibility and consistency of b-ADC and s-ADC, calculated from b-DWI and s-DWI, respectively, were assessed using the intraclass correlation coefficient (ICC). A statistical significance level of *p* < 0.05 was considered. Results: The s-DWI generated by the unsupervised learning framework ADCVCGAN scored significantly higher than a-DWI in quantitative metrics such as PSNR, SSIM, FSIM, MSE, WPSNR, and WMSE, with statistical significance (*p* < 0.001). This performance is comparable to the optimal level achieved by the latest synthetic algorithms. Subjective scores for lesion visibility, image anatomical details, image distortion, and overall image quality were significantly higher for s-DWI and b-DWI compared to a-DWI (*p* < 0.001). At the same time, there was no significant difference between the scores of s-DWI and b-DWI (*p* > 0.05). The consistency of b-ADC and s-ADC readings was comparable among different readers (ICC: b-ADC 0.87–0.90; s-ADC 0.88–0.89, respectively). The repeatability of b-ADC and s-ADC readings by the same reader was also comparable (Reader1 ICC: b-ADC 0.85–0.86, s-ADC 0.85–0.93; Reader2 ICC: b-ADC 0.86–0.87, s-ADC 0.89–0.92, respectively). Conclusions: ADCVCGAN shows excellent promise in generating gastric cancer DWI images. It effectively reduces scanning time, improves image quality, and ensures the authenticity of s-DWI images and their s-ADC values, thus providing a basis for assisting clinical decision making.

## 1. Introduction

Gastric cancer is a prevalent malignant tumor with high morbidity and mortality rates worldwide [1]. Multiparametric magnetic resonance imaging (mp-MRI) is a crucial diagnostic, staging, and prognostic tool for gastric cancer [2]. Diffusion-weighted imaging (DWI), a key component of mp-MRI, indirectly reflects microscopic changes in cell density and tissue structure by measuring the diffusion movement of water molecules [3,4]. Due to the rapid cell proliferation and tight cell arrangement in gastric cancer, the diffusion movement of water molecules is restricted, resulting in high signal intensity on DWI and a lower quantitative measurement value known as the apparent diffusion coefficient (ADC) compared to normal gastric tissue [5]. Recent studies have demonstrated a strong correlation between the ADC value and the TNM stage of gastric cancer, the level of tissue differentiation, the expression of Ki-67, and the assessment of chemotherapy efficacy [6,7,8,9]. Standard DWI utilizes single-shot echo-planar imaging (SS-EPI) to acquire excitation signals across the entire abdominal region. However, this technique is prone to reducing the signal-to-noise ratio of DWI images, aggravating the deformation, and increasing artifacts due to factors such as the inhomogeneous $B_{0}$ field, long acquisition time, eddy currents, and peristaltic movement of the stomach [10,11]. One of the most important parameters in the DWI is the diffusion sensitivity factor b; its increase means the strength of the applied gradient is getting larger. While this enhances the contrast between tumors and normal tissues to a certain extent, it also increases the phase dispersion of water molecules, making the above phenomenon more pronounced. ZOOMit DWI employs a parallel emission technique that only excites signals from the region of interest and images a smaller field of view [12,13]. While this approach can mitigate artifacts and image distortion while improving spatial resolution, it still suffers from a reduced signal-to-noise ratio and challenges in aligning high and low b-value images. Low-quality DWI introduces structural biases in tissue imaging and compromises the accuracy of ADC value calculation and subsequent gastric cancer assessment. Although increasing the number of excitations can enhance the quality of gastric DWI images, excessive excitations prolong the scanning time, compromising patient comfort and leading to artifacts caused by involuntary and physiological movements. Thus, achieving high-quality DWI images while minimizing scanning time presents a significant technical challenge.

In recent years, the emergence of artificial intelligence algorithms has provided innovative solutions for traditional medical image problems. One such algorithm, the cycle generative adversarial network (CycleGAN) [14], demonstrates immense potential in enhancing the quality of medical images without relying on hardware devices. CycleGAN consists of two generators and two discriminators. The generators analyze and establish mappings between unpaired input and reference images for imitation learning, while the discriminators differentiate between generator-generated and natural images. Through a constant game between the generator and discriminator and the application of the cycle consistency loss function, the generator can ultimately generate synthetic images that closely resemble authentic images in terms of quality. Consequently, several variant network frameworks based on CycleGAN have emerged to improve synthetic image generation further. In the field of medical image denoising, researchers such as Gu et al. [15] have proposed a novel, tunable CycleGAN architecture that features a single generator, utilizing adaptive normalization layers to achieve better denoising. Zhou and colleagues introduced CycleWGAN [16], a supervised deep learning model, which was compared with traditional denoising methods on PET images, demonstrating that CycleWGAN achieved the best results. Kwon and their team proposed a novel CycleGAN architecture [17] without loops, consisting of a single generator and discriminator, yet still ensuring cycle consistency. This architecture was tested on a large number of low-dose CT images at various levels, achieving excellent results. In the realm of enhancing medical image resolution, Liu and their colleagues introduced a novel Multi-CycleGAN framework [18], which uses pseudo-cycle consistency modules to control the consistency of generation and domain control modules to provide additional constraints, achieving good results. Liang and their team [19] proposed a novel model that uses unpaired CT and CBCT images in an unsupervised learning environment for image-to-image translation, achieving good results. In the field of medical image artifact removal, Lee and their colleagues [20] proposed a simple and effective unsupervised learning method for MAR, based on a novel CycleGAN architecture derived from optimal transport theory for disentangling in an appropriate feature space, achieving good results in removing metallic artifacts. Yoo and their colleagues [21] proposed a new model based on CycleGAN, aiming to automatically eliminate artifacts in fundus photographs. In the field of multimodal synthesis [22,23,24], variants of the CycleGAN model have also achieved good results. In the enhancement of DWI image quality, Hu and their team [25] designed a CycleGAN-based model that maintains computational precision and repeatability, independent of the MRI scanner, enhancing the quality of prostate DWI images and accelerating the generation of ADC images. Hu and their team [26] designed a new GAN-based model that uses denoising and edge enhancement techniques, enhancing DWI image quality on a multi-center prostate dataset and reducing scan time, achieving good results.

Inspired by this, we envisioned utilizing the CycleGAN-based unsupervised learning framework to construct a mapping between lower b-value (b = 800 ${s/\mathrm{mm}}^{2}$) DWI (a-DWI) and higher b-value (b = 1200 ${s/\mathrm{mm}}^{2}$) DWI (b-DWI), generating the latter by the former, and ultimately obtaining synthetic gastric cancer s-DWI images with good image quality and realism while effectively reducing scanning time.

This study aims to develop and assess an unsupervised learning framework for attention dual contrast vision transformer cyclegan (ADCVCGAN) using a recurrent adversarial generative network. The framework aims to enhance the quality of gastric cancer images, shorten the scanning time for gastric cancer ZOOMit DWI, and evaluate the accuracy of s-DWI and ADC images. The study encompasses five steps: (1) MRI examination, (2) model training, (3) model testing, (4) image quality assessment, and (5) ADC assessment, as depicted in Figure 1.

## 2. Materials and Methods

### 2.1. Patients

The study obtained approval from the local ethical review board. We enrolled 200 gastric cancer patients treated at the First Affiliated Hospital of Nanjing Medical University between February 2022 and May 2023. The study sample consisted of 151 males and 49 females. The inclusion criteria were as follows: (1) patients with a confirmed diagnosis of gastric cancer through gastroscopy prior to MRI examination, (2) availability of complete clinical information and gastric cancer stage diagnosis, (3) patients who had not undergone prior gastric cancer surgery, (4) patients who exhibited good cooperation during the examination, resulting in images of high quality without artifacts such as motion blur or metal implants. Lastly, the gastric cancer patients were randomly divided into a training set of 150 and a test set of 50, adhering to a ratio of 3:1. The patient information under different datasets is shown in Figure 2.

### 2.2. MRI Examinations

Before examining this study, the patients underwent an 8 h fasting period to ensure gastric emptying. Immediately before the mp-MRI scan, the patients consumed 800 mL of warm water to fill the gastric cavity. Ten minutes prior to image acquisition, if there were no contraindications (such as prostatic hypertrophy, glaucoma, or severe heart disease), the patients received an intramuscular injection of 10mg of scopolamine hydrochloride to minimize artifacts caused by gastric peristalsis.

All patients diagnosed with gastric cancer in our study underwent mp-MRI of the abdomen using a 3T MRI scanner (MAGNETOM Skyra, Siemens Healthcare, Erlangen, Germany). The imaging was performed with an 18-channel body phased-array coil and a corresponding 32-channel spine coil. The acquisition sequences included transverse-axial plane T2-FBLADE, T1WI-VIBE-FS, low b-value (b = 50 ${s/\mathrm{mm}}^{2}$) DWI (low-DWI), a-DWI, and b-DWI. The specific scanning sequence setups are shown in Table 1, in which the scanning time for low-DWI, a-DWI, and b-DWI was 3 min 4 s.

### 2.3. Data Preprocessing

Preprocessing precedes model training, which entails the selection and cropping of high-quality DWI (low-DWI, a-DWI, and b-DWI) images, excluding those that fail to encompass gastric cancer, exhibit substantial distortion, or harbor prominent artifacts. This process is then augmented by affine registration, employing an open-source advanced normalization tool (ANTs), to ensure accurate alignment of the DWI images. Subsequently, we selected 5 to 15 DWI images from each patient. In the training set, comprising 150 patients, a total of 1500 DWI images were selected. In the test set, which included 50 patients, 500 DWI images were chosen. All 2D axial slices were rescaled to a uniform resolution of 256 × 256 pixels to facilitate model training.

### 2.4. Model Training

Our proposed unsupervised learning network framework, ADCVCGAN, integrates the U-net framework with the vision transformer (ViT) module [27,28]. Additionally, we incorporate the convolutional block attention module (CBAM) [29] into the generator. We introduce the dual contrast loss [30] in the discriminator, utilizing the b-DWI as a reference to generate high-quality and accurate s-DWIs from the acquired a-DWI.

The original CycleGAN is depicted in Figure 3, where the generator *G* is designed to convert and synthesize the input image *x* into a synthetic image $G_{\left(x\right)}$ that closely resembles the target domain *y* image. However, the original CycleGAN generator has limitations in preserving and conveying global feature information, resulting in poor quality and realism of the generated gastric cancer DWI images. To address this issue, this study incorporates the ViT-based U-net framework into the generator, as illustrated in Figure 4. Firstly, the U-net structure extracts and retains vital features and detailed information specific to gastric cancer tissue. At the same time, skip connections are used to mitigate information loss. Additionally, the transformer’s self-attention mechanism, displayed in Figure 5, is utilized to automatically prioritize information across different locations of the image during the generation of gastric cancer DWI images, enabling a better understanding of the global structure of the gastric cancer tissue image and focusing on areas with finer details, resulting in clearer and more realistic generated images. Finally, the CBAM module is incorporated to extract additional gastric cancer image feature information from both the channel and spatial dimensions.

The objective of the CycleGAN discriminator, or $D_{y}$, is to differentiate authentic images from the target domain y and the synthesized images $G_{\left(x\right)}$ produced by the generator *G*. By utilizing these images, the discriminator learns the distinctive features of the target domain *y*. However, this approach poses a challenge as the discriminator can be easily misled when slight modifications are made to certain features of source domain samples, making them resemble the target domain *y*. Consequently, the discriminator mistakenly identifies the synthesized image $G_{\left(x\right)}$ as genuine, generating DWI gastric cancer images of inadequate quality and realism by the generator *G*. In order to address this issue, our work draws inspiration from Wang et al. [30] and proposes a solution (illustrated in Figure 6) that leverages samples from the source domain.

The discriminator assigns class 1 to authentic images while designating the synthesized images and samples from the source domain as class 0. Furthermore, it incorporates images from the source domain as negative samples. This arrangement compels the discriminator to guide the generator in synthesizing images that deviate from the source domain within the latent space, thereby ensuring the production of high-quality and authentic DWI gastric cancer images.

### 2.5. Experimental Settings

ADCVCGAN adopts a structure similar to CycleGAN, featuring two generators (*G* and *F*) and two discriminators ($D_{x}$ and $D_{y}$). In the ADCVCGAN framework, a U-net architecture based on the ViT module is introduced alongside adding the CBAM module within the generator. Furthermore, the discriminator incorporates the dual contrast loss, leveraging negative samples from the source domain to discern between real and synthetic images. The training process of ADCVCGAN involves five iterations on the generator followed by one iteration on the discriminator. The parameter settings include λ = 10, β = 0.5, *epoch* = 200, and *batch size* = 1.

All algorithms were tested on a Linux system equipped with four NVIDIA Tesla V100 for our experiments. The implementations used Python 3.6 (available at https://www.python.org/downloads/release/python-3615/ accessed on 6 August 2024) and Tensorflow 1.14 (available at https://github.com/tensorflow/tensorflow/releases/tag/v1.14.0 accessed on 6 August 2024).

### 2.6. Image Quality Assessment

To validate the efficacy of the ADCVCGAN model, we fed 50 a-DWI images from the validation set into the model to generate s-DWI images. We quantitatively assessed the image quality using metrics such as the peak signal-to-noise ratio (PSNR), mean square error (MSE), weighted PSNR (WPSNR), weighted MSE (WMSE), structural similarity (SSIM), and feature similarity (FSIM). PSNR was used to evaluate the signal-to-noise ratio of a-DWI, s-DWI, and b-DWI images. MSE reflected the overall differences between these images. WPSNR, a weighted version of PSNR, incorporated the diagnostic expertise of senior clinical doctors by assigning higher weights to lesions of clinical concern and using a weighted matrix to accurately reflect the signal-to-noise ratio. WMSE similarly used a weighted matrix to reflect the overall differences. SSIM and FSIM, respectively, evaluated the structural and feature similarities between a-DWI, s-DWI, and b-DWI images. Furthermore, we selected the latest methods in the medical image synthesis field, based on CycleGAN and its variants, as a benchmark to verify the superiority of our model’s performance.

A subjective assessment of DWI images was conducted by two radiologists, each with extensive experience (8 and 10 years) in abdominal MRI diagnostics, using a Siemens workstation. In a blinded evaluation, the evaluators, without access to individual patient details, including lesion location and endoscopic pathology outcomes, rated a composite set of three DWI modalities (a-DWI, b-DWI, and s-DWI) at three distinct time points, separated by two-week intervals, in random order. The assessment encompassed four critical criteria: lesion severity, anatomical precision, image integrity, and overall image quality, employing a Likert scale (ranging from 1 for the poorest to 5 for the highest quality). Their evaluations were systematically compared to T2-FBLADE and T1WI-VIBE-FS images for reference.

### 2.7. ADC Measurement Assessment

In order to evaluate the consistency and repeatability of b-ADC and s-ADC, this study employed a two-point (b = 50 ${s/\mathrm{mm}}^{2}$ and b = 1200 ${s/\mathrm{mm}}^{2}$) method to calculate them separately. Firstly, a highly experienced physician with over 20 years of expertise in diagnostic abdominal imaging acted as a coordinator. The coordinator identified the precise location of the nodule on the ADC image, referring to the imaging report and other MRI serial images. Subsequently, the coordinator selected the most representative ADC level with the largest nodule area. Next, two radiologists with a decade of experience in abdominal MRI independently drew a circular region of interest (ROI) in the center of the lesion. This was performed three times for each lesion without knowledge of the clinical, surgical, and histological findings. The mean ADC value of each ROI was recorded. After a month, the ADC images were shuffled, and the two radiologists repeated the ROI outlining the procedure. The initial measurements provided by the two radiologists were used to evaluate the consistency of b-ADC and s-ADC. The subsequent measurements were used to assess the reproducibility of b-ADC and s-ADC.

### 2.8. Statistical Analyses

This study underwent statistical analysis using MedCalc software version 22.032. An independent samples *t*-test was employed to compare the differences of continuous variables that followed a normal distribution. At the same time, the Mann-Whitney *U*-test was utilized for continuous variables that did not conform to a normal distribution. The Wilcoxon signed-rank test was conducted to compare the quality scores of the DWI. The intraclass correlation coefficient (ICC) was implemented to evaluate the repeatability of ADC readings by the same reader, and it was also utilized to assess the consistency of ADC readings by different readers.

### 2.9. Ethical Statement

We confirmed that all methods were carried out in accordance with relevant guidelines and regulations, and informed consent for patients was waived by the Research Ethics Committee of the Nanjing Medical University. All experimental protocols and data in this study were approved by the Research Ethics Committee of the Nanjing Medical University. Approval number: NMUE2021301.

## 3. Results

### 3.1. Demographic Characteristics

There was no significant difference in the mean age of the training and test sets (mean age: 62 ± 11 years vs. 61 ± 13 years, *p* = 0.968).

### 3.2. Image Quality Assessments Results and Analyses

We present the DWI images of six representative patients with gastric cancer from the test set, as depicted in Figure 7. The location of the gastric cancer lesions is outlined with a red solid line. Visually, the shape of the gastric cancer lesions, the noise distribution, and the boundaries definition in the s-DWI images closely resemble those in the b-DWI images, without any apparent anatomical deformations. Regarding suppressing internal signals, primarily water, in the stomach (indicated by blue arrows), the signal intensity distribution in s-DWI aligns well with that in b-DWI. It differs significantly from a-DWI, leading to better highlighting of the lesions. In the comprehensive characterization of proximal and distal lymph nodes (indicated by green arrows), s-DWI is consistent with both a-DWI and b-DWI, providing accurate representation without the loss of additional details. Significantly, our observations revealed that s-DWI offered superior local details and border definition in comparison to b-DWI for gastric cancer patients with small lymph nodes and no hemangiomas (e.g., patients A and B). Conversely, for patients with large lymph nodes (e.g., patients C, D, and E), s-DWI exhibited high similarity to b-DWI. Likewise, for patients with small lesions and no lymph nodes (e.g., patient F), both s-DWI and b-DWI demonstrated high similarity.

The quantitative evaluation results were consistent with visual perception, as shown in violin Figure 8. The s-DWI exhibits superior performance to a-DWI in terms of the distributions, medians, and quartile ranges of PSNR, MSE, WPSNR, WMSE, SSIM, and FSIM scores. Specifically, the average PSNR values for s-DWI and a-DWI are 30.4467 and 22.9009, respectively, with average MSE values of 0.000882 and 0.0056, respectively. The average WPSNR for s-DWI is 30.4967, and the average WMSE for s-DWI is 0.00088064. The average SSIM and FSIM values for s-DWI and a-DWI are 0.9366 and 0.8456, respectively, and 0.8653 and 0.793, respectively. These differences are statistically significant (*p* < 0.001), suggesting that s-DWI is visually comparable to the reference b-DWI image in terms of image quality, outperforming a-DWI. Furthermore, the observation that the WPSNR of s-DWI exceeds the PSNR and the WMSE is less than the MSE indicates that significant portions of the weighting matrix, corresponding to the lesion regions chosen by clinical experts, exhibit reduced error, thereby reinforcing the effectiveness of the synthesized effect. Table 2 presents a quantitative comparison of our method against several advanced comparative algorithms. The experimental findings demonstrate that our method surpasses the latest algorithms in various metrics evaluations, achieving outstanding results.

The subjective quality scores of the three groups of DWI images, evaluated by the two reviewers, are presented in Table 3. Both b-DWI and s-DWI obtained higher scores than a-DWI regarding the visibility of gastric cancer foci, depiction of anatomical details, absence of image distortion, and overall image quality. These differences were statistically significant (*p* < 0.001). However, no statistically significant differences were observed between b-DWI and s-DWI regarding the visibility of gastric cancer lesions, depiction of anatomical details, image distortion, and overall image quality score (*p* > 0.05). This observation suggests that s-DWI closely approximates the reference image, b-DWI, in terms of subjective clinical scoring, demonstrating high authenticity.

### 3.3. Ablation Studies for ADCVCGAN

To thoroughly assess the efficacy of each module, we employed CycleGAN as our backbone and conducted a module stacking analysis to gauge their individual impact on the overall performance. Abbreviations used are: Vision Transformer (ViT), Convolutional Block Attention Module (CBAM), and Dual Contrast Loss (DCL). The quantitative results of our ablation study are detailed in Table 4.

The experimental outcomes demonstrate the collective impact of each module on the overall performance. ViT, through its self-attention mechanism, facilitates the model’s ability to automatically discern distinct regions in DWI images, thereby enhancing its comprehension of gastric cancer tissue’s global structure and refining details, thereby enhancing image fidelity. The CBAM module effectively extracts a broader range of gastric cancer image characteristics from both channel and spatial aspects, leading to clearer generated images. DCL classifies real images as Class 1, synthetic images and source domain samples as Class 0, and incorporates source domain images as negative examples, compelling the discriminator to guide the generator to produce images in the latent space that deviate further from the source domain, ensuring the genuineness of the generated DWI images. Comparative analysis of the evaluation metrics, including PSNR, MSE, WPSNR, WMSE, SSIM, and FSIM, reveals that ViT outperforms CBAM and DCL, with a more substantial improvement in model efficiency.

### 3.4. ADC Measurement Assessment Results and Analyses

We present ADC images of six patients with gastric cancer, illustrated in Figure 9, and delineate the location of gastric cancer lesions using a red solid line. Table 5 demonstrates the concordance assessment of b-ADC and s-ADC between two readers. The numerical results indicate that both b-ADC and s-ADC exhibit high and comparable concordance (ICC: b-ADC 0.87–0.90; s-ADC 0.88–0.89, respectively) across different tissues. Table 6 showcases the reproducibility of b-ADC and s-ADC for the same reader. The numerical results reveal that both b-ADC and s-ADC possess high reproducibility and are comparable across different tissues. For reader 1, the ICC is 0.85–0.86 for b-ADC and 0.85–0.93 for s-ADC. For reader 2, the ICC is 0.86–0.87 for b-ADC and 0.89–0.92 for s-ADC. These findings suggest that s-ADC shares highly similar clinical characteristics with the reference image b-ADC and can partially replace it.

## 4. Discussion

The main contribution of this study is the proposal of the unsupervised learning framework ADCVCGAN, which effectively reduces scanning time without the need for additional hardware devices. Simultaneously, it improves the quality of DWI images and ensures image authenticity. The reliability of the framework is evaluated based on both computer vision and clinical value. DWI is a widely used MRI technique crucial in diagnosing, staging, and monitoring gastric cancer. The quality of DWI images significantly impacts their clinical evaluation effectiveness and the accuracy of the generated ADC values. By maintaining high-quality DWI images, hospitals can save time and cost and reduce the probability of patient involuntary movement caused by long scanning times. This improvement in image quality enhances the diagnostic efficacy of DWI and ADC for gastric cancer.

Due to its ability to simulate data distribution and image transformation, several scholars have explored using CycleGAN and its extension algorithms to establish connections between different images and address challenges encountered in clinical settings. For instance, Yan et al. [35] constructed a denoising network based on CycleGAN to generate high-quality CT scan images from low-quality ones, enabling the evaluation of CT scan quality. Dong et al. [36] developed a CycleGAN-based denoising network to transform cone-beam computed tomography (CBCT) scans with artifacts into more apparent plan computed tomography (PCT) scans, thereby reducing artifact presence in CBCT images. Sun et al. [37] employed an MD-CycleGAN model to extract artifact-free images from CT scans. Moreover, Wang et al. [30] proposed the DC-CycleGAN, a dual-contrast model based on CycleGAN, to synthesize MRI images with fewer artifacts from CT images. These applications aim to reduce patient scanning radiation and time. Despite the potential of CycleGAN-based models in solving traditional medical image problems, their adoption in clinical practice still needs to be improved. One reason is that the generated images, although visually high-quality, need more clinical significance as they differ from authentic scanned images. Additionally, previous studies have primarily evaluated the generated images using traditional computer vision evaluation methods, leading to skepticism among clinicians regarding their value. To address these concerns, this study assesses the quality and clinical significance of s-DWI and its s-ADC using traditional visual evaluation indexes, such as PSNR, MSE, WPSNR, WMSE, SSIM and FSIM, and clinical evaluation methods.

The study introduces the ViT-based U-net framework into the generator of CycleGAN to address the information loss problem and improve the generation of DWI images for gastric cancer. The U-net structure extracts and preserves vital features and detailed information about gastric cancer tissue, while the jump connection mitigates information loss. Additionally, the self-attention mechanism of the transformer selectively focuses on different regions of the image to better understand the overall structure and highlight specific areas, resulting in clearer and more realistic generated images. To further enhance the extraction of gastric cancer image features, the CBAM module is incorporated, analyzing both channel and spatial dimensions. Finally, negative samples from the source domain are injected to guide the generator in synthesizing images that differ from the source domain in the latent space, ensuring high-quality and realistic generated DWI images for gastric cancer. From the macroscopic visualization, it can be observed that s-DWI exhibited similar characteristics to b-DWI in terms of the shape of gastric cancer lesions, noise distribution, and border clarity, without any noticeable anatomical deformation. The suppression of signal intensity within the stomach was also highly comparable between s-DWI and b-DWI. In terms of the detailed characterization of both proximal and distal lymph nodes, the findings from s-DWI aligned with those from a-DWI and b-DWI, accurately reflecting the actual situation and not resulting in any loss of detail. Furthermore, for gastric cancer patients with small lymph nodes, the local details of the lesion and the border clarity were superior in s-DWI compared to b-DWI. Conversely, for patients with large lymph nodes, the overall image quality of s-DWI and b-DWI was highly similar. Lastly, for patients with small lesions and no lymph nodes, the overall image quality of s-DWI and b-DWI was also found to be very similar. These findings indicate that the proposed unsupervised learning framework ADCVCGAN successfully focuses on gastric cancer features at different scales, comprehends local and global structures, prioritizes detailed regions, and produces s-DWI images that closely resemble b-DWI. The quantitative results are consistent with the visual evaluation, demonstrating that s-DWI surpasses a-DWI in image quality based on the distribution, median, and interquartile range of PSNR, MSE, WPSNR, WMSE, SSIM and FSIM scores. Moreover, s-DWI exhibits similar quality to the reference image b-DWI and surpasses a-DWI in clinical subjective quality scores, including the visibility of gastric cancer lesions, anatomical image details, image distortion, and overall image quality. The difference between the scores of b-DWI and s-DWI is not statistically significant, further emphasizing the close similarity of s-DWI to b-DWI and its high clinical authenticity. Additional comprehensive ablation studies, as presented in Table 4, were carried out. The results consistently revealed that our proposed module contributed to improved model performance, with ViT demonstrating superior efficacy compared to both CBAM and DCL. This superiority is primarily attributed to ViT’s self-attention mechanism, which facilitates automatic focus on diverse image regions during the generation of DWI data. This enhanced capacity to grasp the global tissue structure and concentrate on intricate details ultimately boosts image quality and authenticity. This study also confirms the reproducibility and consistency of s-ADC and b-ADC, demonstrating that s-ADC shares clinical properties highly similar to the reference image b-ADC. Through computer vision and clinical value assessment, the proposed unsupervised learning framework ADCVCGAN effectively reduces scanning time, emphasizes the details of gastric cancer lesions without relying on additional hardware devices or scanning time, and enhances DWI image quality while ensuring image authenticity.

There are several limitations to this study. Firstly, only b-DWI was utilized as the standard b-value in this experiment and the validity of this model for other b-values requires further verification. Secondly, as DWI and ADC values can vary depending on factors such as field strength, equipment from different manufacturers, and diverse populations, the generalizability of this model still needs confirmation through large-scale, multi-center experiments. Lastly, due to the limited sample size and absence of related studies, this investigation did not categorize the tumors based on malignancy or benignity. Future studies should focus on evaluating the discriminatory capability of s-ADC in distinguishing benign and malignant tumors.

## 5. Conclusions

This paper introduces an unsupervised learning framework, ADCVCGAN, and investigates its potential for enhancing DWI image quality and reducing scan time. To assess the model’s performance, comprehensive experiments were conducted, and comparisons were made with the latest synthesis algorithms. The results demonstrate that the proposed model outperforms the state-of-the-art in objective metrics such as PSNR, MSE, WPSNR, WMSE, SSIM, and FSIM, and also achieves favorable subjective evaluation results. Additionally, ablation studies confirm that the modules proposed in this paper enhance the model’s overall performance, with ViT showing more significant improvements over CBAM and DCL. This is primarily because ViT can automatically focus on different parts of the image during DWI image generation, thereby better understanding the global structure of gastric cancer tissue images and focusing on more detailed areas, ensuring the quality and authenticity of the generated images. Although the research findings suggest that the proposed model performs well in the synthesis task, this paper did not test with b-DWI as the standard b-value, and further investigation is needed to determine the model’s applicability to more b-value DWI. The model’s generalization capabilities, as well as its ability to distinguish between benign and malignant gastric tumors in s-DWI and s-ADC, warrant further study in future work.

## Figures and Tables

**Figure 1 bioengineering-11-00805-f001:**
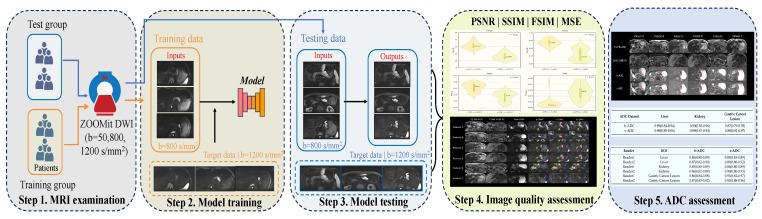
Presents the overall learning flowchart consisting of five steps. In Step 1, all patients underwent mp-MRI using b-values of 50, 800, and 1200 ${s/\mathrm{mm}}^{2}$. For Step 2, model training was conducted with b = 800 ${s/\mathrm{mm}}^{2}$ as the input data from the training group, and b = 1200 ${s/\mathrm{mm}}^{2}$ as the target data. In Step 3, gastric cancer images were synthesized using the model inputs of b = 800 ${s/\mathrm{mm}}^{2}$ from the test group. Step 4 involved assessing the quality of the synthesized gastric cancer images through metrics such as the peak signal-to-noise ratio, structural similarity, feature similarity, mean square error, and subjective reading score by diagnosticians. Finally, Step 5 focused on analyzing the ADC consistency and repeatability of the synthetic gastric cancer images.

**Figure 2 bioengineering-11-00805-f002:**
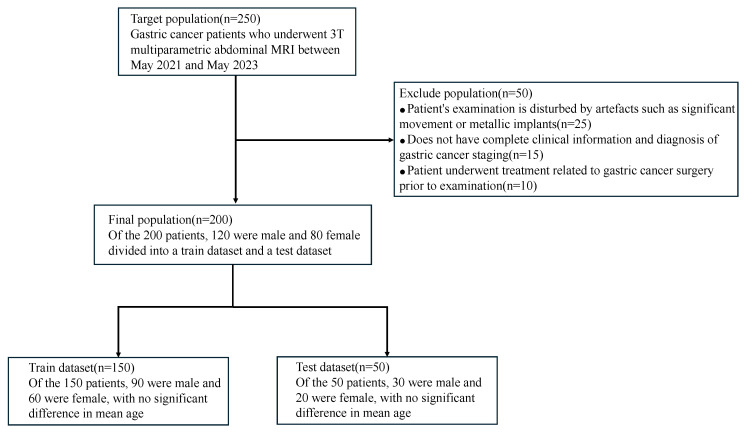
Patient infographic with different datasets. This includes number of patients and gender.

**Figure 3 bioengineering-11-00805-f003:**
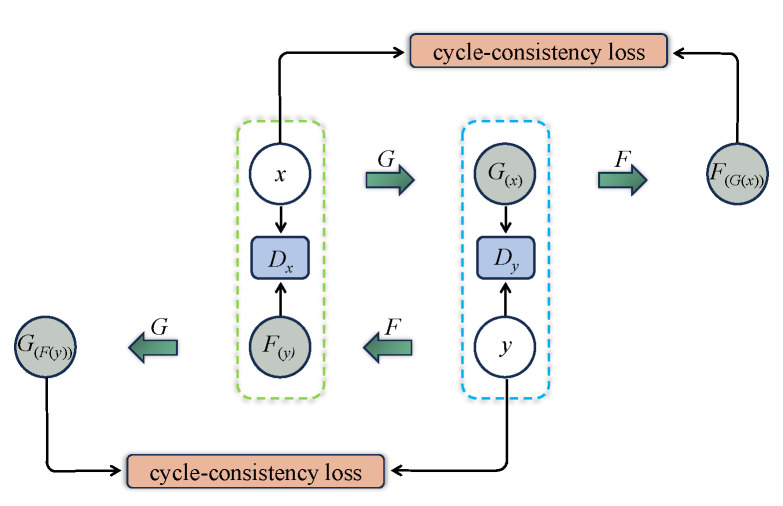
Schematic of the original CycleGAN model. *G* and *F* stand for generators, $D_{x}$ and $D_{y}$ stand for discriminators, $G_{\left(x\right)}$ stands for the image of *x* generated by generator *G*, $F_{\left(y\right)}$ stands for the image of *y* generated by generator *F*, $G_{\left(F(y)\right)}$ stands for the image of $F_{\left(y\right)}$ generated by generator *G*, and $F_{\left(G(x)\right)}$ stands for the image of $G_{\left(x\right)}$ generated by generator *F*.

**Figure 4 bioengineering-11-00805-f004:**
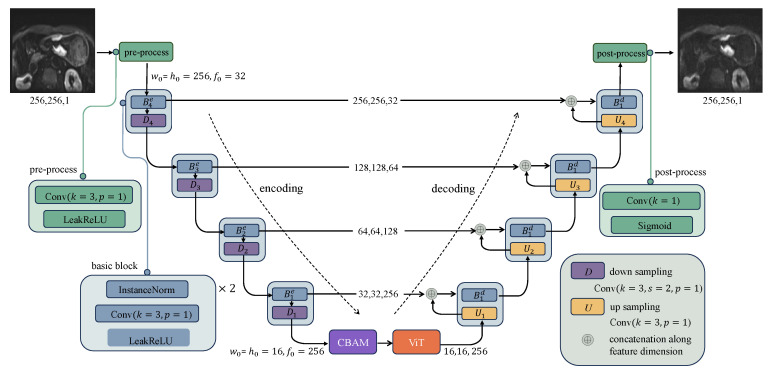
ADCVCGAN model generator section.The coding path of U-net extracts features from the input through four layers of convolution and downsampling, and passes the extracted features from each layer to the corresponding layer of the decoding path through skip connections. In the encoding path of U-net, the preprocessing layer converts the image into a tensor with dimensions (*$w_{0}$,$h_{0}$,$f_{0}$*), and the preprocessed tensor halves the width $w_{0}$ and the height $h_{0}$ in each downsampled block while the feature dimension $f_{0}$ is doubled.

**Figure 5 bioengineering-11-00805-f005:**
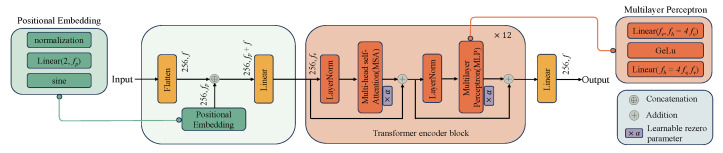
ViT module in ADCVCGAN.ViT is composed primarily of a stack of transformer encoder blocks. To construct an input to the stack, the ViT first flattens an encoded image along the spatial dimensions to form a sequence of tokens. The token sequence has length *w* × *h*, and each token in the sequence is a vector of length *f*. It then concatenates each token with its two-dimensional Fourier positional embedding of dimension $f_{p}$ and linearly maps the result to have dimension $f_{v}$. To improve the Transformer convergence, we adopt the rezero regularization scheme and introduce a trainable scaling parameter α that modulates the magnitudes of the nontrivial branches of the residual blocks. The output from the transformer stack is linearly projected back to have dimension *f* and unflattened to have width *w* and *h*. In this study, we use 12 transform encoder blocks.

**Figure 6 bioengineering-11-00805-f006:**
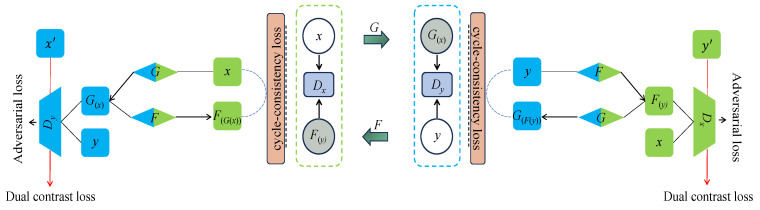
The structure of dual contrast [30]. The introduction of images from the source domain as negative samples compels the discriminator to steer the generator towards synthesizing images that diverge from the source domain within the latent space. Here, $x^{\prime }$ and $y^{\prime }$ represent randomly selected negative samples from the source image.

**Figure 7 bioengineering-11-00805-f007:**
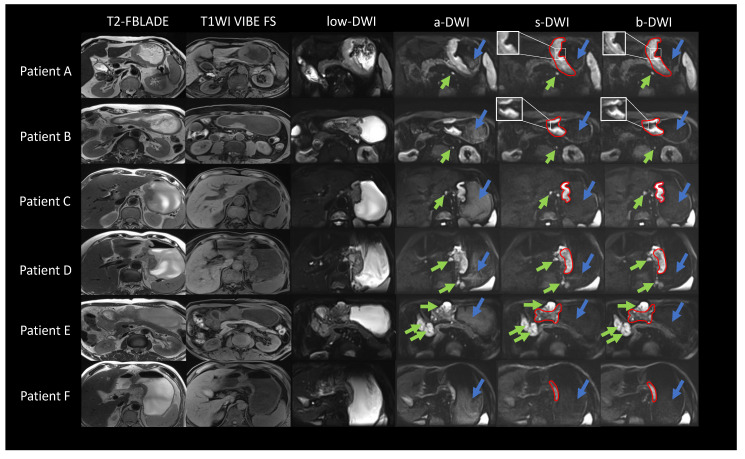
DWI images of patients with gastric cancer. The figure shows the lesion range indicated by the red solid line, the signal inside the stomach indicated by the blue arrow, and the lymph node indicated by the green arrow. Patient A, male, 74 years old, subcardia-gastric body lesser curvature-gastric angle occupancy, infiltrating ulcer type. Patient B, male, 64 years old, occupancy of the gastric antrum, infiltrating ulcer type. Patient C, male, 66 years old, cardia-gastric lesser curvature occupancy, infiltrating ulcer type. Patient D, female, 72 years old, cardia to the lesser curvature of the gastric body occupation, infiltrating ulcer type. Patient E, male, 57 years old, gastric angle-sinus occupation, infiltrating ulcerative type. Patient F, male, 72 years old, lateral to the lesser curvature of the gastric body, infiltrating ulcer type.

**Figure 8 bioengineering-11-00805-f008:**
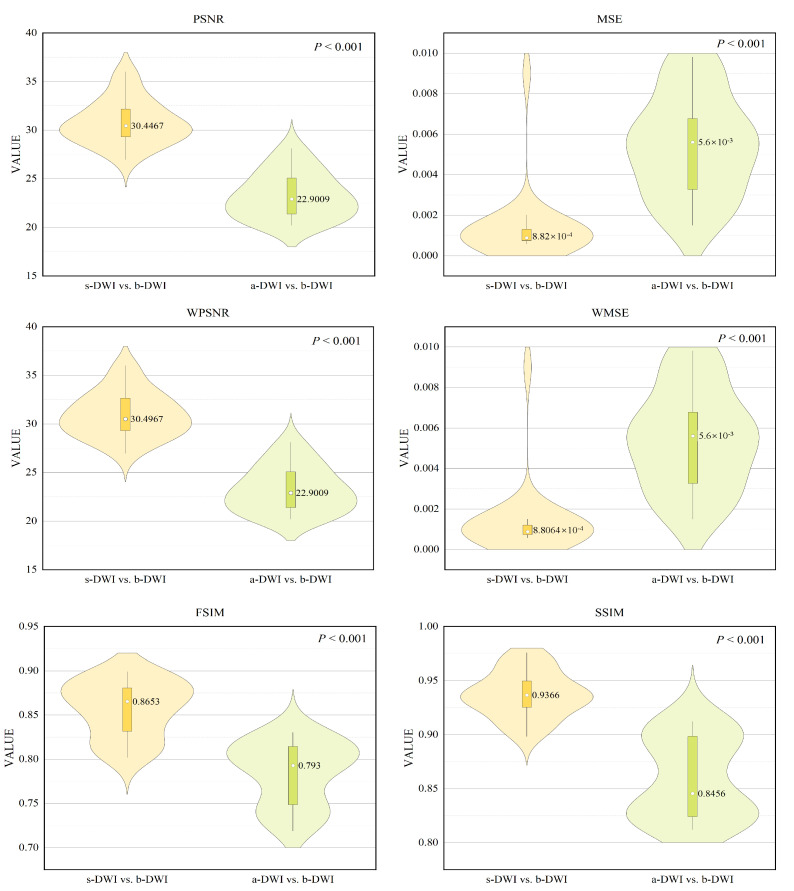
Violin plots of the quantitative metric distributions of the DWI.

**Figure 9 bioengineering-11-00805-f009:**
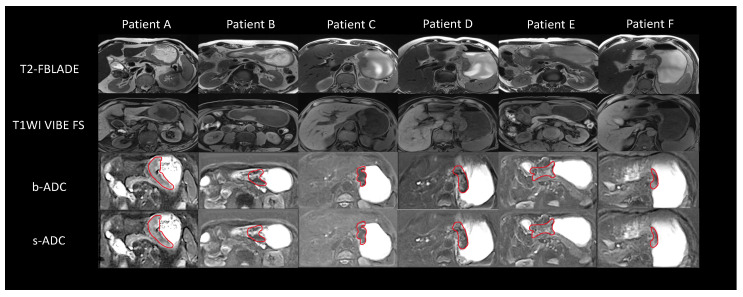
ADC images of patients with gastric cancer.

**Table 1 bioengineering-11-00805-t001:** MRI scanning sequence parameters.

Parameter	T2-FBLADE	T1WI-VIBE-FS	low-DWI	a-DWI	b-DWI
Field-of-view FOV (${\mathrm{mm}}^{2}$)	340×340	320×260	250×125	250×125	250×125
Imaging matrix	320×320	240×195	192×96	192×96	192×96
Thickness (mm)	3	3	4	4	4
Distance fact	0.6	0	0	0	0
B-value (${s/\mathrm{mm}}^{2}$)	n.a.	n.a.	50	800	1200
Number of averages	1	1	2	4	6
Echo time (ms)	98	1.3	80	80	80
Repetition time (ms)	4100	3.31	3500	3500	3500
Flip angle	99	9	90	90	90
Bandwidth (Hz/pixel)	780	445	1490	1490	1490
Scan time (min)	1 min 48 s	13s			

**Table 2 bioengineering-11-00805-t002:** Comparison of metrics of different methods on our dataset.

Methods	PSNR ↑	MSE ↓	WPSNR ↑	WMSE ↓	SSIM ↑	FSIM ↑
CycleGAN [31]	27.0158	0.001347	27.0164	0.001338	0.9012	0.8302
AttentionGAN [32]	27.2902	0.001133	27.2956	0.001129	0.9123	0.8352
RegGAN [33]	28.2411	0.001004	28.2488	0.000988	0.9249	0.8587
ADCycleGAN [34]	29.4879	0.000962	29.5231	0.000951	0.9355	0.8620
Ours	30.4467	0.000882	30.4502	0.000861	0.9366	0.8653

**Table 3 bioengineering-11-00805-t003:** Subjective quality ratings of the three sets of DWI images by two readers.

Reader	Evaluation Metrics	DWI Image Quality Score			*p*-Value		
		a-DWI	b-DWI	s-DWI	a-DWI vs. b-DWI	a-DWI vs. s-DWI	b-DWI vs. s-DWI
Reader1	lesion visibility	2.06 ± 0.745	4.18 ± 0.735	4.15 ± 0.842	<0.001	<0.001	0.554
Reader2		2.03 ± 0.755	3.96 ± 0.777	4.03 ± 0.792	<0.001	<0.001	0.184
Reader1	anatomical details	1.98 ± 0.795	3.89 ± 0.863	3.85 ± 0.742	<0.001	<0.001	0.343
Reader2		1.92 ± 0.752	3.91 ± 0.757	3.93 ± 0.859	<0.001	<0.001	0.210
Reader1	image distortion	2.02 ± 0.845	3.94 ± 0.786	3.92 ± 0.820	<0.001	<0.001	0.367
Reader2		2.08 ± 0.795	4.00 ± 0.772	4.06 ± 0.712	<0.001	<0.001	0.318
Reader1	overall quality	2.21 ± 0.768	4.23 ± 0.742	4.26 ± 0.733	<0.001	<0.001	0.634
Reader2		2.06 ± 0.879	4.15 ± 0.777	4.15 ± 0.833	<0.001	<0.001	0.998

**Table 4 bioengineering-11-00805-t004:** Quantitative results of CycleGAN-based ablation experiments.

Methods	PSNR ↑	MSE ↓	WPSNR ↑	WMSE ↓	SSIM ↑	FSIM ↑
CycleGAN	27.0158	0.001347	27.0164	0.001338	0.9012	0.8302
CycleGAN-ViT	28.4533	0.000977	28.4551	0.000963	0.9302	0.8554
CycleGAN-CBAM	27.5579	0.001158	27.5591	0.001150	0.9105	0.8427
CycleGAN-DCL	27.2201	0.001233	27.2211	0.001228	0.9088	0.8346
CycleGAN-ViT-CBAM	29.4589	0.000899	29.4598	0.000884	0.9351	0.8613
CycleGAN-ViT-DCL	29.1203	0.000945	29.1221	0.000932	0.9344	0.8601
CycleGAN-CBAM-DCL	28.2301	0.001024	28.2320	0.001001	0.9245	0.8521
Ours	30.4467	0.000882	30.4502	0.000861	0.9366	0.8653

**Table 5 bioengineering-11-00805-t005:** Reading consistency between two readers.

ADC Dataset	Liver	Kidney	Gastric Cancer Lesion
b-ADC	0.89 (0.84–0.94)	0.90 (0.83–0.96)	0.87 (0.79–0.93)
s-ADC	0.89 (0.85–0.96)	0.89 (0.81–0.94)	0.88 (0.81–0.97)

**Table 6 bioengineering-11-00805-t006:** Repeatability of reading by the same reader.

Reader	ROI	b-ADC	s-ADC
Reader1	Liver	0.86 (0.80–0.89)	0.85 (0.83–0.89)
Reader2	Liver	0.87 (0.82–0.90)	0.89 (0.80–0.92)
Reader1	Kidney	0.85 (0.80–0.89)	0.86 (0.80–0.89)
Reader2	Kidney	0.86 (0.82–0.89)	0.90 (0.80–0.93)
Reader1	Gastric Cancer Lesion	0.86 (0.84–0.88)	0.93 (0.84–0.97)
Reader2	Gastric Cancer Lesion	0.87 (0.83–0.92)	0.92 (0.88–0.96)

## Data Availability

We thank the Department of Radiology, Jiangsu Provincial People’s Hospital, our collaborating hospital, for providing private datasets, which are available upon request from the corresponding author.

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
