# Peer review of "Synthesizing High b-Value Diffusion-Weighted Imaging of Gastric Cancer Using an Improved Vision Transformer CycleGAN"

_bioengineering, 2024, doi:10.3390/bioengineering11080805_

Round 1

Reviewer 1 Report

Comments and Suggestions for Authors

This manuscript reports the outcomes of a systematic approach to the synthesis of high b-value diffusion weighted images using a new variant of cycleGAN. The methodology presented is well explained and logical and the framework used has demonstrated useful improvements in image quality and authenticity with a minimum of scanning time. The manuscript is well written, and the carefully thought-out figures complement and summarise the discussion and the chosen topic is of definite relevance to bioengineering. There are some points that I think the authors should address and these are detailed below.

1.      I think that the coverage of the literature in the introduction is not quite as thorough as it ought to be. There is omission of some relevant literature on the use of GAN type methods to improve b-values in imaging of human tissue other than for gastric cancer (e.g. Hu et al. Radiology Artificial Intelligence 2021 3(5) e200237, which looks at this sort of analysis applied to the prostate). The authors should recheck and reconsider their literature review to ensure appropriate coverage.

2.      The term ‘b-value’ is used in the manuscript without being properly defined. Whilst I accept that this term is in widespread use in imaging, there are many readers who would not be familiar with this parameter and so properly defining what it means in the manuscript is essential.

3.      Figure 2 caption. This caption does not make sense as worded. It appears that the figure is really about the inclusion of patients in the different datasets for the study. Reword the caption to make it more informative.

4.      Figure 3. The associated caption should clearly define the different parameters shown in the schematic so that the figure is comprehensible independently of the main text.

Comments on the Quality of English Language

The quality of the English in this paper is generally quite good with just some minor errors that a careful editorial check would pick up.

Reviewer 2 Report

Comments and Suggestions for Authors

The following points need to be addressed in the revision

Abstract

  1. The abstract is not adequate in length. It should be summarized by removing the introductory text or information irrelevant to highlight in the abstract.
  2. A lot of text, related to the patients’ data is never required in the abstract.
  3. The main innovation and contribution of this research should be clarified in the abstract.
  4. Comparison with SOTA techniques should be mentioned in the abstract.
  5. Can you compare the results with similar studies or some previous findings of similar studies?

Introduction

  1. The motivation of the study is not focused and needs improvement in the revised version.
  2. The text in Figure 1 is small the quality of the image needs improvement by selecting the color contrast to make it improved for the audience.
  3. The image quality assessment should also be measured using WPSNR. Add its values along with other measures.

Methodology/Results/Conclusions

  1. I could not find the results of the statistical analysis (Section 2.8) being referred to. Add these results.
  2. The conclusions section is too brief and must be improved to improve the conclusive ideas and key points therein the article.

Round 2

Reviewer 2 Report

Comments and Suggestions for Authors

The reference numbers, figure numbers, etc. are missing indicated by "?" in the revised manuscript. Correct it.